# Comparative transcriptomics reveal differential gene expression among *Plasmodium vivax* geographical isolates and implications on erythrocyte invasion mechanisms

Daniel Kepple[1]*, Colby T. Ford[2,3], Jonathan Williams[1], Beka Abagero[1], Shaoyu Li[4], Jean Popovici[5], Delenasaw Yewhalaw[6,7], Eugenia Lo[1,8]*

1 Biological Sciences, University of North Carolina, Charlotte, North Carolina, United States of America, 2 Bioinformatics and Genomics, University of North Carolina, Charlotte, North Carolina, United States of America, 3 School of Data Science, University of North Carolina, Charlotte, North Carolina, United States of America, 4 Mathematics and Statistics, University of North Carolina, Charlotte, North Carolina, United States of America, 5 Malaria Research Unit, Institut Pasteur du Cambodge, Phnom Penh, Cambodia, 6 Tropical and Infectious Diseases Research Center, Jimma University, Jimma, Ethiopia, 7 School of Medical Laboratory Sciences, Faculty of Health Sciences, Jimma University, Jimma, Ethiopia, 8 Microbiology and Immunology, College of Medicine, Drexel University, Philadelphia, Pennsylvania, United States of America

* dkepple@uncc.edu (DK); el855@drexel.edu (EL)

**Data Availability Statement:** Sequences for the 10 Ethiopian transcriptomes are available on the

## Abstract

The documentation of *Plasmodium vivax* malaria across Africa especially in regions where Duffy negatives are dominant suggests possibly alternative erythrocyte invasion mechanisms. While the transcriptomes of the Southeast Asian and South American *P. vivax* are well documented, the gene expression profile of *P. vivax* in Africa is unclear. In this study, we examined the expression of 4,404 gene transcripts belong to 12 functional groups and 43 erythrocyte binding gene candidates in Ethiopian isolates and compared them with the Cambodian and Brazilian *P. vivax* transcriptomes. Overall, there were 10–26% differences in the gene expression profile amongst geographical isolates, with the Ethiopian and Cambodian *P. vivax* being most similar. Majority of the gene transcripts involved in protein transportation, housekeeping, and host interaction were highly transcribed in the Ethiopian isolates. Members of the reticulocyte binding protein *PvRBP*2a and *PvRBP*3 expressed six-fold higher than Duffy binding protein *PvDBP*1 and 60-fold higher than *PvEBP*/*DBP*2 in the Ethiopian isolates. Other genes including *PvMSP3.8*, *PvMSP3.9*, *PvTRAG2*, *PvTRAG14*, and *PvTRAG22* also showed relatively high expression. Differential expression patterns were observed among geographical isolates, e.g., *PvDBP*1 and *PvEBP*/*DBP*2 were highly expressed in the Cambodian but not the Brazilian and Ethiopian isolates, whereas *PvRBP*2a and *PvRBP*2b showed higher expression in the Ethiopian and Cambodian than the Brazilian isolates. Compared to *Pvs*25, gametocyte genes including *PvAP2-G*, *PvGAP* (female gametocytes), and *Pvs*47 (male gametocytes) were highly expressed across geographical samples.

National Center for Biotechnology Information
Short Read Archive under BioProject:
PRJNA784582. All code is available on GitHub at
https://github.com/colbyford/vivax_transcriptome_
comparisons.

**Funding:** This research was funded by National
Institutes of Health R01 AI162947 to E.L and R01
AI173171 to E.L. and J.P. The funders had no role
in study design, data collection and analysis,
decision to publish, or preparation of the
manuscript.

**Competing interests:** The authors have declared
that no competing interests exist.

## Author summary

*Plasmodium vivax* malaria is a neglected tropical disease, despite being more geographically widespread than any other form of malaria and causes 132–391 million clinical infections each year. The documentation of *P. vivax* infections in different parts of Africa where Duffy-negative individuals, who were previously thought to be immune to *P. vivax* malaria, are dominant suggested that there are alternative pathways for *P. vivax* to invade human erythrocytes. Experimental approaches to unveil parasite invasion ligands are greatly limited due to a lack of reliable long-term culturing techniques and thus remains largely unexplored. Findings of this study are the first to examine the transcriptomes of African *P. vivax* and compare such to other geographical isolates with the goal to provide an important baseline for future comparisons of *P. vivax* transcriptomes in Duffy-negative infections. Our analyses also highlight potential biomarkers for improved gametocyte detection to better monitor the spread of *P. vivax* malaria.

## 1. Introduction

*Plasmodium vivax* Duffy binding protein (*PvDBP1*), which binds to the cysteine-rich region II of the human glycoprotein Duffy Antigen-Chemokine Receptor (DARC) [1–3], was previously thought to be the exclusive invasion mechanism for *P. vivax* [4]. However, the several reports of *P. vivax* infections in majority Duffy-negative countries [3] have raised important questions of how *P. vivax* invades erythrocytes. It was previously hypothesized that either mutations in *PvDBP*1 or a weakened expression of DARC in Duffy-negative individuals allowed *P. vivax* invasion in Duffy-negative erythrocytes [5,6] and thus enabled *P vivax* to spread in Africa. Despite mutational differences observed in *PvDBP*1 between Duffy-positive and Duffy-negative infections, these differences do not lead to binding of Duffy-negative erythrocytes [4] and suggested alternative invasion pathways.

 *Plasmodium vivax* is closely related to a large clade of malaria parasites that infect lesser apes and ceropithecoids (old world monkeys) of Southeast Asia [7,8]. The exact origin of human *P. vivax* is still heavily debated, with evidence of *P. vivax* originating in Africa [7] and in Asia [9,10] both being supported. The first reference genome of *P. vivax* was Salvador I, isolated from *Saimiri boliviensis* monkeys in El Salvador in 2008 [11], followed by the P01 genome isolated from a *P. vivax* patient in Indonesia in 2016 [12]. The *P. vivax* nuclear genome is 29 megabases with a 39.8% G-C composition and 6,642 genes distributed amongst 14 chromosomes [12]. Several large gene subfamilies have been identified in the P01 genome, including the most abundant *Plasmodium* interspersed repeat (*pir*; formally described as *vir*) genes in the subtelomeric region, followed by unclassified *Plasmodium* exported proteins and tryptophan-rich antigen proteins [12]. Remarkably, across the genome, approximately 77% of genes are orthologous between *P. falciparum*, *P. knowlesi*, and *P. yoelii* [11]. Genes involved in key metabolic pathways, housekeeping functions, and membrane transporters are highly conserved between *P. vivax* and *P. falciparum* [11]. However, *P. vivax* isolates from Africa, Southeast Asia, South America, and Pacific Oceania have significantly higher nucleotide diversity at the genome level compared to *P. falciparum* [13,14], likely due to variations in transmission intensity, frequency of gene flow via human movement, age of host-pathogen interactions, and host susceptibility [15].

 Genes such as erythrocyte binding protein (*PvEBP*), reticulocyte binding protein (*PvRBP*), merozoite surface protein (*PvMSP*), apical membrane antigen 1 (*PvAMA1*), anchored

micronemal antigen (*PvGAMA*), Rhoptry neck protein (*PvRON*), and tryptophan-rich antigen genes (*PvTRAg*) families are suggested to play a role in erythrocyte invasion [13,16], especially in low-density infections [17–21]. Prior genomic studies have shown high polymorphisms in genes such as *PvDBP*1, *PvMSP*1, *PvMSP*7, and *PvRBP*2c [22–26]. Erythrocyte binding protein gene, *PvEBP*, a paralog of PvDBP1, harbors all the hallmarks of a *Plasmodium* red blood cell invasion protein. *PvEBP* is similar to *PcyM DBP*2 sequences in *P. cynomolgi* and contains a Duffy-binding like domain [27]. Binding assay of *PvEBP* region II (171–484) showed moderate binding activity to Duffy-negative erythrocytes [4]. Both *PvDBP*1 and *PvEBP* (*PvDBP*2) genes exhibit high genetic diversity and are common antibody binding targets associated with clinical protection [28,29]. Host receptors for both *PvRBP*1b and *PvRBP*1a proteins remain undetermined, but several members of *PvRBP*2 (*PvRBP*2a, *PvRBP*2b, *PvRBP*2c, *PvRBP*2d, *PvRBP*2e, *PvRBP*2p1, and *PvRBP*2p2) are orthologous to *PfRH*2a, *PcyRBP*2, and *PfRH*2b, with *PvRBP*2a and *PfRh*5 share high structural similarity [30,31]. *PvRBP*2b and *PvRBP*2c are orthologous to *PcyRBP*2b and *PcyRBP*2c, respectively [32]. The receptor for PvRBP2a was previously identified as CD98, a type II transmembrane protein that links to one of several L-type amino acid transporters to form heterodimeric neutral amino acid transport systems [33]; the receptor for PvRBP2b is transferrin receptor 1 (TfR1) [34]. The PvRBP2b-TfR1 interaction plays a critical role in reticulocyte invasion in Duffy-positive infections [34]. MSP1 also shows a strong binding affinity, with high-activity binding peptides (HABPs) clustered close to these two fragments at positions 280–719 and 1060–1599, respectively [35], suggesting a critical role in erythrocyte invasion. Although the *MSP*7 gene family shows no binding potential, it forms a complex with *PvTRAg*36.6 and *PvTRAg*56.2 on the surface, likely for stabilization purposes at the merozoite surface [20]. A comparison of *P. vivax* transcriptomes between *Aotus* and *Saimiri* monkeys indicated that the expression of six *PvTRAg* genes in *Saimiri P. vivax* was 37-fold higher than in the *Aotus* monkey strains [19], five of which bind to human erythrocytes [20,36]. Although most TRAg receptors remain poorly characterized and unnamed, the receptor of PvTRAg38 has been identified as Band 3 [37].

Recent advance in short-term *in vitro* culturing and schizont-enrichment methodologies have enabled transcriptomic sequencing of *P. vivax* enabling a comprehensive review of stage-specific gene expression profile and structure, of which thousands of splices and unannotated untranslated regions were characterized [38]. The transcriptomes of Cambodian [39] and Brazilian [40] *P. vivax* field isolates showed high expression levels and large populational variation amongst host-interaction transcripts. For example, the MSP1 gene family was highly upregulated in the Cambodian *P. vivax* compared to the Brazilian ones. Similar trends were also observed in *PvDBP*1, *PvEBP*, *PvMA*, *PvRA*, *PvRBP*2a, *PvMSP*5, and *PvMSP*4, highlighting geographical differences in the gene expression profile. In *P. falciparum*, distinct phenotypic and expression levels of erythrocyte binding antigen (EBA) and reticulocyte binding-like homologue (Rh) gene families were observed among geographical isolates due to varying immunogenic pressures [41]. Heterogeneity of gene expression has been documented amongst *P falciparum*-infected samples, implying that the parasites can modulate the gene transcription process through epigenetic regulation [42]. However, the transcriptomic profile of African *P. vivax* remains unexplored, and it is unclear if there is heterogeneity among the continental isolates. In addition, our previous study found that two CPW-WPC genes PVP01_0904300 and PVP01_1119500 expressed in the male gametocytes, and *Pvs230* (PVP01_0415800) and *ULG*8 (PVP01_1452800) expressed in the female gametocytes were highly expressed relative to *Pvs25* in the Ethiopian *P. vivax* [43]. While these genes have a potential to be used for gametocyte detection, it remains unclear if such expressional patterns are similar in other geographical isolates.

In this study, we aimed to 1) examine the overall gene expression profile of 10 Ethiopian *P. vivax* with respect to different intraerythrocytic lifecycle stages; 2) determine the expression

levels of previously characterized erythrocyte binding gene candidates [13]; 3) compare gene expression profiles of the Ethiopian *P. vivax* with the Cambodian [39] and Brazilian [40] isolates from *in vitro* especially on the erythrocyte binding and male/female gametocyte gene candidates. These findings are the first to describe *P. vivax* transcriptomes from East Africa and provide critical insights into alternative parasite invasion ligand proteins other than PvDBP1. A systematic comparison of gene expression profiles among the African, Southeast Asian, and South American isolates will deepen our understanding of *P. vivax* transcriptional machinery and invasion mechanisms.

## 2. Materials and methods

### 2.1 Ethics statement

Scientific and ethical clearance was obtained from the institutional scientific and ethical review boards of Jimma University, Ethiopia (#03-246-796-22) and University of North Carolina at Charlotte, USA (IRBIS-21-0371). Written informed consent/assent for study participation was obtained from all consenting heads of households, parents/guardians (for minors under 18 years old), and individuals who were willing to participate in the study.

### 2.2 Sample preparation

Ten microscopy-confirmed *P. vivax* samples were collected from Duffy positive patients at hospitals in Jimma, Ethiopia. These patients had 4,000 parasites/μL parasitemia and had not received prior antimalarial treatment. A total of 10mL whole blood was preserved in sodium heparin tubes at the time of collection. Red blood cell pellets were isolated and cryo-preserved with two times glycerolyte 57 and stored in liquid nitrogen within one hour of collection. Prior to culture, samples were thawed by adding 0.2V of 12% NaCl solution drop-by-drop followed by a 5-minute room temperature incubation. Ten-times volume of 1.6% NaCl solution was then added drop-by-drop to the mixture and the samples were centrifuged at 1000 rcf for 10 minutes to isolate the red blood cell pellet. This process was repeated with a 10x volume of 0.9% NaCl. Following centrifugation, the supernatant was removed via aspiration, and 18mL of sterile IMDM (also containing 2.5% human AB plasma, 2.5% HEPES buffer, 2% hypoxanthine, 0.25% albumax, and 0.2% gentamycin) per 1mL cryo-preserved cell mixture was added to each sample for a final hematocrit of 2%. 10% Giemsa thick microscopy slides were made to determine the majority parasite stage and duration of incubation required; being 20–22 hours for the majority trophozoites and 40–44 hours for the majority ring to ensure samples were majority schizont for future analysis. Samples were incubated at 37°C in a 5% O2, 5% CO2 with the same infected patient blood *in situ* to allow maturation and minimize potential culturing effects of the transcriptome. *In vitro* maturation was validated through microscopic smears 20–40 hours after the initial starting time, dependent on the majority stage. To minimize oxidative stress, each culture was checked more than two times and returned to a 5% oxygen environment immediately after checking.

Cultured pellets were isolated via centrifugation and placed in 10x volume trizol for RNA extraction. RNA extraction was performed using direct-zol RNA prep kit according to the manufacturer's protocol, followed by two rounds of DNA digestion using the DNA-free kit (Zymo). Samples were analyzed with a nanodrop 2000 and RNA Qubit to ensure sample concentrations were above 150 ng total for library construction. For samples with no significant amount of DNA or protein contaminants, RNA libraries were constructed using Illumina rRNA depletion library kits according to the manufacturer's protocol. Completed libraries were quality checked using a bioanalyzer to ensure adequate cDNA was produced before sequencing. Sample reads were obtained using Illumina HiSeq 2x150bp configuration to

obtain at least 35 million reads per sample. Sequence reads were aligned with HISAT2 [44], using the Rhisat2 R package [45] to the P01 *P. vivax* reference genome and all human reads were filtered out using SAMtools [46] (implemented in the R package [47]). The alignment was mapped to the P01 reference annotation using the Rsubread package [48].

### 2.3 Data analyses

To further confirm samples were majority schizont stage, sequence reads of each sample were deconvoluted in CIBERSORTx [49] based on *P. berghei* homologs [50]. We used the published matrix to determine the frequency of expression for each gene calculated for rings, trophozoites, and schizonts, respectively. Transcripts that were expressed 30% or more were sorted into their respective stages. All reads were annotated using the Rsubread package and classified into 12 different categories by function. We then examined the top 30 transcribed genes using the counts per million (CPM) metric.

Our previously published whole genome sequence data identified several mutations and structural polymorphisms in genes from the *PvEBP*, *PvRBP*, *PvMSP*, and *PvTRAg* gene families that are likely to involve in erythrocyte invasion [13]. Specific binding regions in some of the genes such as *PvDBP*1, *PvEBP/DBP*2, *PvRBP*2b, and *PvMSP*3 have been identified [51]. To further explore the putative function, we compared relative expression levels of 43 erythrocyte binding gene candidates (S1 Table) in the 10 Ethiopian *P. vivax* samples with other geographical isolates that were of majority schizont stage. We used the CPM and TPM (transcripts per million) metrics in R package edgeR [52]. The CPM metric was used to obtain the top 30 transcripts overall and does not consider gene length, while TPM considers gene length for normalization and allows an unbiased conclusion to be made relative between and to other transcriptomes [40]. We then transformed the data using $log(2)$TPM +1 to illustrate relative expression levels via a heat map with an average abundance. We also selected 25 gametocyte gene candidates, 15 of which were shown to correlate to female gametocyte development and nine to male gametocytes [43,53], to assess their expression levels relative to the standard *Pvs*25 in the samples. In addition, we examined the expression of AP2-G that is a critical transcription factor for both male and female gametocyte development [54].

### 2.4 Comparison of datasets

Previously published, raw RNA-seq data of four *in vitro* Cambodian [39] and two *in vitro* Brazilian [40] *P. vivax* samples were downloaded from the GitHub repositories and analyzed with the same bioinformatic methods described above to minimize potential batch effects. The Ethiopian *P. vivax* samples were cultured and sequenced using the same media and timelines (being majority schizont prior to RNA collection) as the Cambodian [39] and Brazilian [40] isolates. To further ensure comparisons are accurate and unbiased, we deconvoluted the parasite stages using the same matrix and found no statistical difference in the average stage composition. We then obtained the average expression and standard deviation in TPM for each gene target and determined potential difference in transcription levels by conducting pairwise differential expression (DE) analysis among the Cambodian, Brazilian, and Ethiopian samples. The expression level of 6,829 genes were examined for DE by edgeR dream [52,55] and variancePartition [56], with adjusted *p*-value<1.0e-6 for DE gene concordance. A linear mixed effects models was used to ensure accuracy in triplicated Brazilian samples, and the Kenward-Roger method was used to estimate the effective degree of freedom for hypothesis testing due to small sample sizes.

## 3. Results

### 3.1 Overview of the Ethiopian *P. vivax* transcriptomes

Based on deconvolution, all 10 Ethiopian *P. vivax* samples had similar proportions of trophozoite and schizont stage (Fig 1A). Only less than 1% of the sequence reads belong to the ring stage. Microscopic results corroborated the deconvolution analyses showing similar proportion of parasite stages in a subset of samples (Fig 1B). The deconvolution of *P. vivax* sequence reads from the Cambodian and Brazilian samples also showed no significant difference in the proportions of trophozoites or schizonts ($P>0.05$; Fig 1A).

Overall, about 64% (4,404 out of 6,830) of the genes were detected with transcription in the Ethiopian *P. vivax*. Of the 4,404 genes, 69% (2,997) were annotated with known functions and 31% (1,407 genes) remain uncharacterized (Fig 2A). We normalized each sample expression profile to TPM to remove technical bias in the sequences and ensure gene expressions were directly comparable within and between samples. Of the 2,997 genes with known function, 21.7% are responsible for housekeeping, and 14.2% genes for post-translation modifications (PTMs) and regulation. The PIR proteins account for 4.8% (212) of all the identified genes and ~2.8% of the genes are involved in host-pathogen interactions. Nearly 52% of all detectable transcripts (2,288 genes) were expressed at a threshold of 20 TPM or above, which were considered as highly transcribed (Fig 2B). These highly transcribed transcripts showed similar proportions of gene categories including unknown, PTM/regulatory, DNA regulation, replication/elongation, host interactions, cell signaling, and resistance. Only transcripts involved in transport and housekeeping showed a slight increase of 2.9% and 1.48%, respectively, indicating a higher activity relative to the other categories. By contrast, transcripts involved in RNA regulation, PIR, and ribosomal activity showed a slight decrease of 2.19%, 1.79%, and 1.71%, indicating an overall lower activity compared to other categories (Fig 2B).

### 3.2 Top 30 transcripts of Ethiopian *P. vivax*

For the 10 Ethiopian *P. vivax* transcriptomes, four genes including PVP01_1000200 (PIR protein), PVP01_0202900 (18s rRNA), PVP01_0319600 (RNA-binding protein), and PVP01_0319500

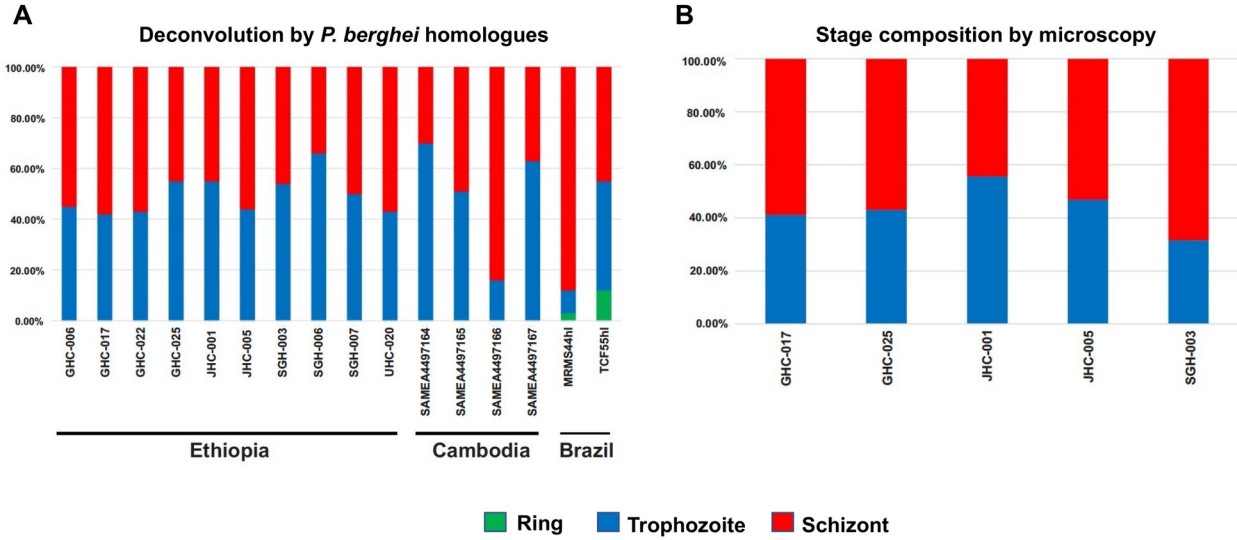

**Fig 1.** (A) CIBERSORTx deconvolution of the 10 Ethiopian, four Cambodian, and two Brazilian *P. vivax* transcriptomes using a *P. berghei* homologue matrix. No significant difference was observed in the proportion of trophozoites and schizonts amongst the isolates ($p>0.05$). (B) Parasite stage based on microscopic analysis of five Ethiopian *P. vivax* samples. No significant difference was observed between microscopy and computational deconvolution for these samples ($p>0.05$).

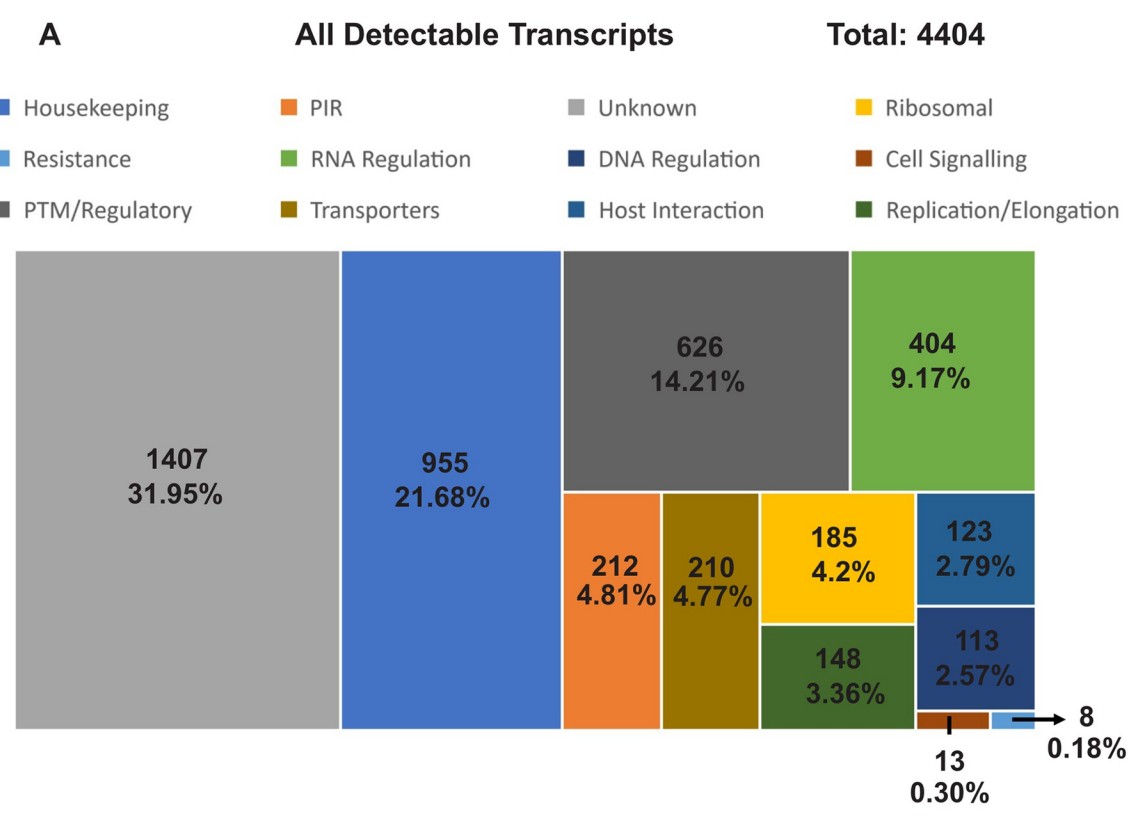

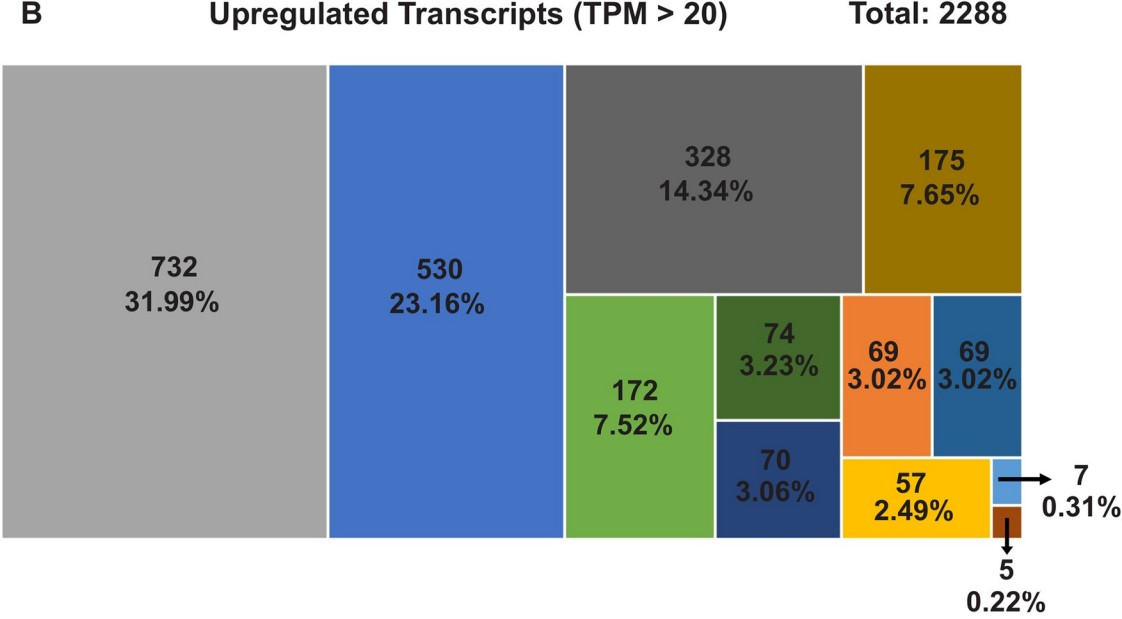

**Fig 2.** Categorization of (A) all detectable transcripts and (B) upregulated (TPM > 20) transcripts for the Ethiopian *P. vivax* by gene function. The numbers shown represent the number of transcripts along with the overall percentage compared to all detected transcripts. Transcripts that were not detected were removed from the analysis. Only transcripts involved in transport and housekeeping showed a slight increase of 2.9% and 1.48%, respectively in the number of upregulated transcripts, indicating a higher activity relative to the other categories. By contrast, transcripts involved in RNA regulation, PIR, and ribosomal activity showed a slight decrease of 2.19%, 1.79%, and 1.71%, indicating an overall lower activity compared to other categories.

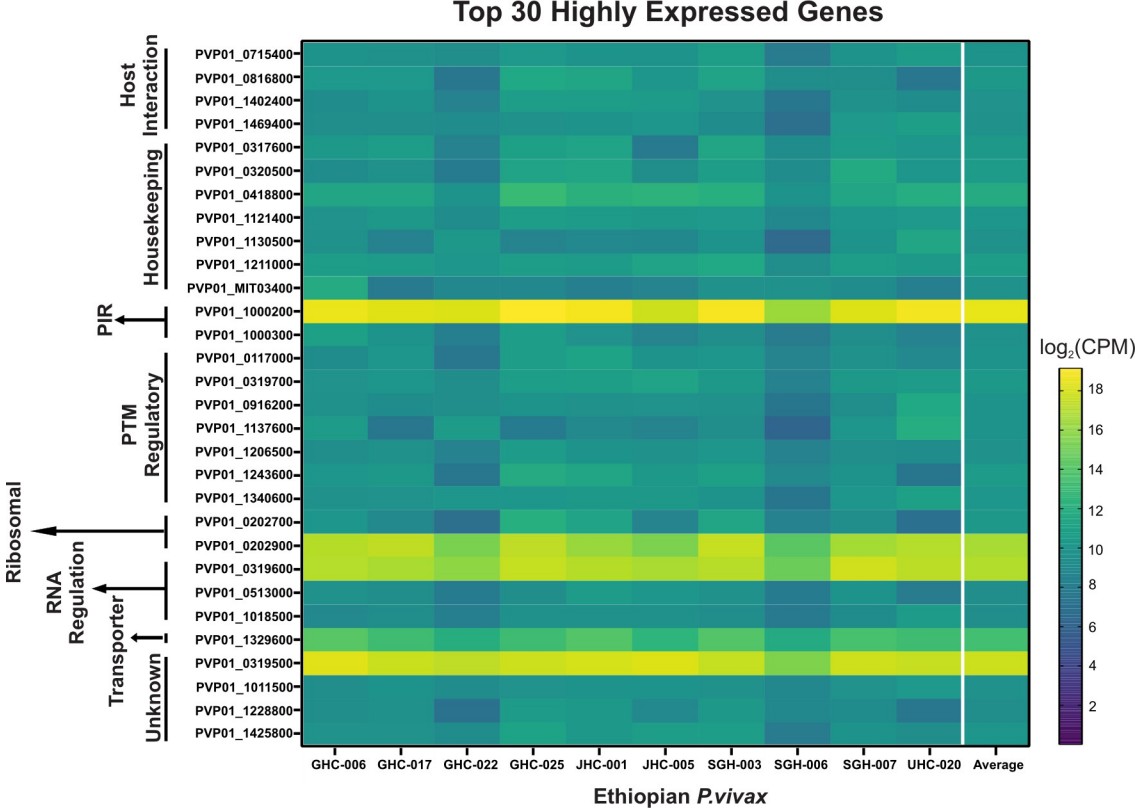

**Fig 3. Heat map showing the top 30 highly transcribed genes based on *log*(2)CPM+1.** Genes are arranged by different functions as indicated on the y-axis. Overall, four genes including PVP01_1000200 (PIR protein), PVP01_0202900 (18s rRNA), PVP01_0319600 (RNA-binding protein), and PVP01_0319500 (unknown function) from four different functional groups were shown to be most highly expressed among the others. Of interest, PVP01_0715400 (merozoite organizing protein), PVP01_0816800 (protein RIPR), PVP01_1402400 (reticulocyte binding protein 2a), and PVP01_1469400 (reticulocyte binding protein 3) were among the top 30 highly expressed genes involved in host interactions.

(unknown function) were the most highly expressed among the others (Fig 3). Transcripts involved in housekeeping and PTM regulation each account for 23.3% of the top 30 highly expressed genes. Among genes involved in host-interactions, PVP01_0715400 (merozoite organizing protein), PVP01_0816800 (protein RIPR), PVP01_1402400 (reticulocyte binding protein 2a), and PVP01_1469400 (reticulocyte binding protein 3) are highly expressed. Five gene transcripts including PVP01_1000200 from the PIR family, PVP01_0319500 of unknown function, PVP01_0202900 a 18S rRNA, PVP01_1329600 a putative glutathione S-transferase, and PVP01_0418800 a putative pentafunctional AROM polypeptide showed most variable expression levels among the 10 samples, with a standard deviation of 20,000 and higher CPM (Fig 3). Three other genes including PVP01_0202700 (28S ribosomal RNA), PVP01_1137600 (basal complex transmembrane protein 1), PVP01_1243600 (replication factor C subunit 3) showed moderate variation ranging from 1,397 to 1,033 CPM. All other genes such as PVP01_1206500 (elongation factor Tu) and PVP01_1011500 (an unclassified protein) showed consistent expression level with variation under 1,000 CPM among samples (Fig 3).

### 3.3 Differentially expressed genes among geographical *P. vivax*

The overall gene expression profile was similar between the Ethiopian and Cambodian *P. vivax*, but different from the Brazilian ones (Fig 4A and S2 Table). Several genes involved in DNA

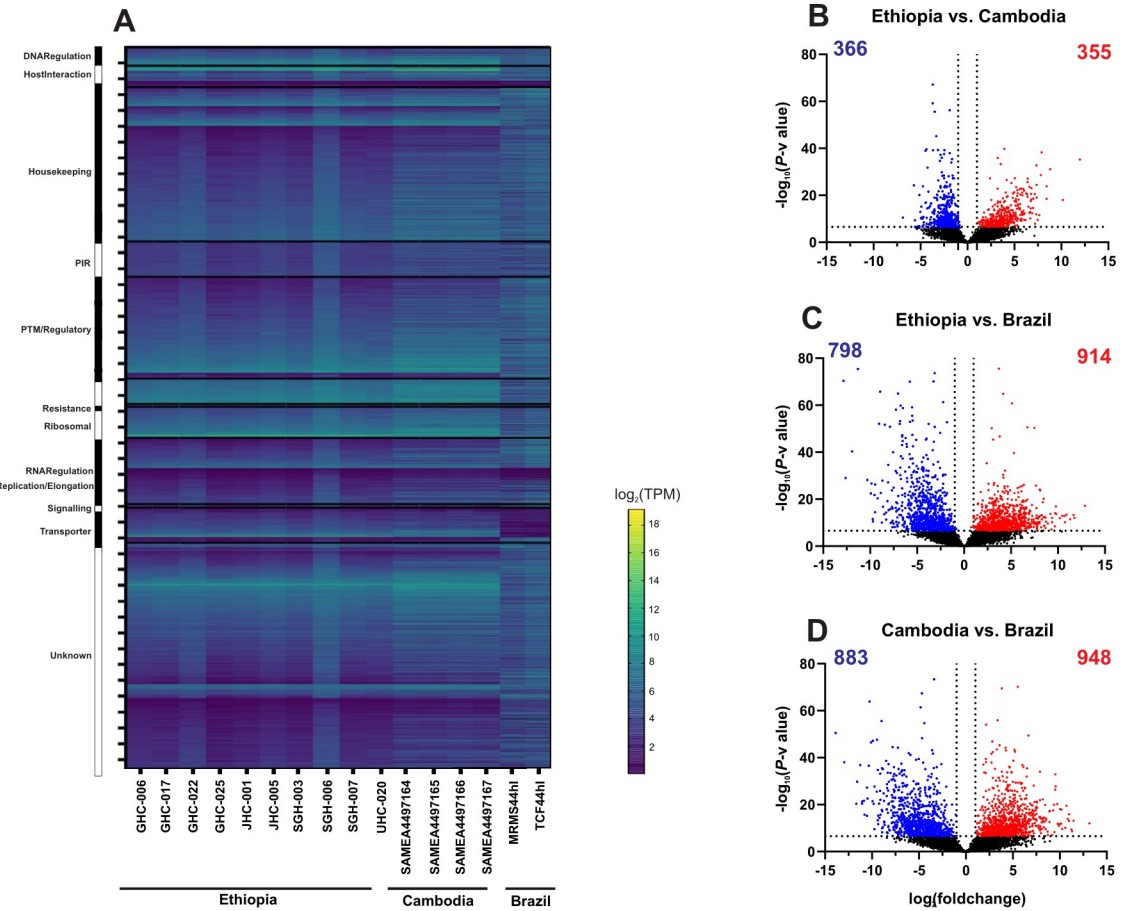

**Fig 4.** (A) Comparisons of the entire transcriptomes with genes sorted by functionality among the Ethiopian, Cambodian, and Brazilian *P. vivax*. The overall gene expression profile was nearly identical between the Ethiopian and Cambodian *P. vivax*, but different from the Brazilian isolates. Several genes involved in DNA regulation, host-interactions, replication, ribosomal, and transportation were upregulated in the Ethiopian and Cambodian isolates but showed considerable downregulation in Brazilian ones. (B-D) Volcano plots based on the Kenward-Roger DE analyses comparing differentially expressed genes between the (B) Ethiopian and Cambodian; (C) Ethiopian and Brazilian; (D) Cambodian and Brazilian isolates. Blue dots represent single genes that are downregulated in the comparison while red dots represent upregulated genes by comparison. About 10% of the detectable transcripts were differentially expressed between the Ethiopian and Cambodian *P. vivax*, but about 25% and 27% variations were detected between the Ethiopian and Brazilian as well as the Cambodian and Brazilian *P. vivax*, respectively. Overall, the Brazilian isolates had more genes that were upregulated compared to the Ethiopian and Cambodian ones.

regulation, host-interactions, replication, ribosomal, and transportation were upregulated in the Ethiopian and Cambodian isolates but showed considerable downregulation in Brazilian ones. Based on the Kenward-Roger DE analyses, a total of 1,831 differentially expressed genes were detected between the Cambodian and Brazilian isolates (CvB), 1,716 between the Ethiopian and Brazilian (EvB), and 721 between the Ethiopian and Cambodian (EvC) isolates (Fig 4B–4D). The EvC analysis showed the lowest differentiation with only 10.6% of the entire transcriptome (Fig 4B), while EvB and CvB showed a greater differentiation of 25.1% and 26.8%, respectively (Fig 4C and 4D). For the 721 genes that were differentially expressed between the Cambodian and Ethiopian *P. vivax*, nearly half of them were significantly upregulated in Ethiopia compared to Cambodia (Fig 4B). Four genes including PVP01_0208700 (V-type proton ATPase subunit C), PVP01_0102800 (chitinase), PVP01_0404000 (PIR protein), and PVP01_0808300 (zinc finger (CCCH type protein) showed low levels of transcription ($\log_{10}P$-value>50; Fig 4B) compared to other DE genes. By contrast, two genes including PVP01_1329600 (glutathione S-

transferase) and PVP01_MIT03400 (cytochrome b) were highly transcribed ($\log_2$fold change>10). For the 1,716 genes that were differentially expressed between the Ethiopian and Brazilian *P. vivax*, 914 of them were highly transcribed (Fig 3C). Of these, three genes including PVP01_1412800 (M1-family alanyl aminopeptidase), PVP01_0723900 (protein phosphatase-beta), and PVP01_0504500 (28S ribosomal RNA) showed a $\log_{10}P$-value greater then 75, indicating substantial expressional differences. For the 1,831 genes that were differentially expressed between the Cambodian and Brazilian *P. vivax*, 948 of them were highly transcribed (Fig 4D). Four genes including PVP01_1005900 (ATP-dependent RNA helicase DDX41), PVP01_0318700 (tRNAHis guanylyltransferase), PVP01_1334600 (60S ribosomal protein L10), and PVP01_1125300 (SURP domain-containing protein) showed substantial expressional differences with $\log_{10}P$-value greater than 75. Two genes, PVP01_0010550 (28S ribosomal RNA) and PVP01_0422600 (early transcribed membrane protein), were shown with low expression ($log_{10}$fold change<-12), while one gene PVP01_0901000 (PIR protein) with substantial expression ($log_{10}$fold change>12). These comparisons further demonstrated the differences in transcriptional patterns between geographical isolates.

### 3.4 Expression of genes related to erythrocyte invasion

Of the 43 candidate genes associated with erythrocyte binding function, *PvDBP*1 on average showed about 10-fold higher expression than *PvEBP/DBP*2, which showed very low expression in four of the Ethiopian *P. vivax* samples (Fig 5). *PvRBP*2b showed four-fold higher expression than *PvEBP/DBP*2, but 50% less than *PvDBP*1. *PvRBP*2a showed consistently the highest expression across all samples, with about 6-fold, 67-fold, and 15-fold higher expression than *PvDBP*1, *PvEBP/DBP*2, and *PvRBP*2b, respectively. Other genes including *PvMSP*3.8, *PvTRAg*14, and *PvTRAg*22 also showed higher expression than *PvDBP*1. Of the 15 *PvTRAg* genes, only *PvTRAg*14 and *PvTRAg*22 showed expression higher than *PvDBP*1; *PvTRAg*23 and *PvTRAg*24 showed the lowest expression. Other putatively functional ligands including *PvRA* and *PvRON4* showed 7–10 times lower expression compared to *PvDBP*1, though *PvGAMA*, *PvRhopH*3, *PvAMA*1, and *PvRON*2 were expressed higher than *PvEBP/DBP*2.

We further compared the expressional pattern of these 43 genes in the Ethiopian *P. vivax* with the Cambodian and Brazilian isolates (Fig 6). Members of the *PvDBP* and *PvRBP* gene family showed generally higher expression in the Cambodian *P. vivax* than the other isolates (Fig 6A). For instance, the expression of *PvDBP*1, *PvRBP*1a, and *PvRBP*1b were significantly higher in the Cambodian than the other isolates ($P$<0.01), whereas *PvRBP*2a and *PvRBP*2b showed higher expression in the Ethiopian *P. vivax* than the others. Compared to the *PvDBP* and *PvRBP* gene families, the expression patterns of *PvMSP* were different (Fig 6B). Most of the *MSP* gene members including *PvMSP*3.5, *PvMSP*3.11, and *PvMSP*4 showed substantially higher expression in the Brazilian *P. vivax* than the other isolates ($P$<0.01). Only *PvMSP*3.8 of the 12 *PvMSP* genes was expressed significantly higher in the Ethiopian than the others ($P$<0.01; Fig 6B). Of the 16 *PvTRAg* genes, *PvTRAg*14 and *PvTRAg*22 showed significantly higher expression in the Ethiopian isolates compared to the others ($P$<0.05; Fig 6C). Eight other members including *PvTRAg*2b, *PvTRAg*7, *PvTRAg*19, *PvTRAg*20, *PvTRAg*21, *PvTRAg*23, *PvTRAg*24, and *PvTRAg*38 showed significantly higher expression in the Brazilian isolates than the others ($P$<0.05; Fig 6C). The remaining nine putatively functional ligands showed relatively similar expression levels, except for *PvMA*, *PvRhopH*3, and *PvTrx-mero* that were highly expressed in the Brazilian isolates ($P$<0.05; Fig 6D).

### 3.5 Expression of female and male gametocyte genes

Based on the expression level of *Pvs25* (PvP01_0616100), all 10 Ethiopian *P. vivax* samples contained submicroscopic gametocytes, in addition to the four samples from Cambodia

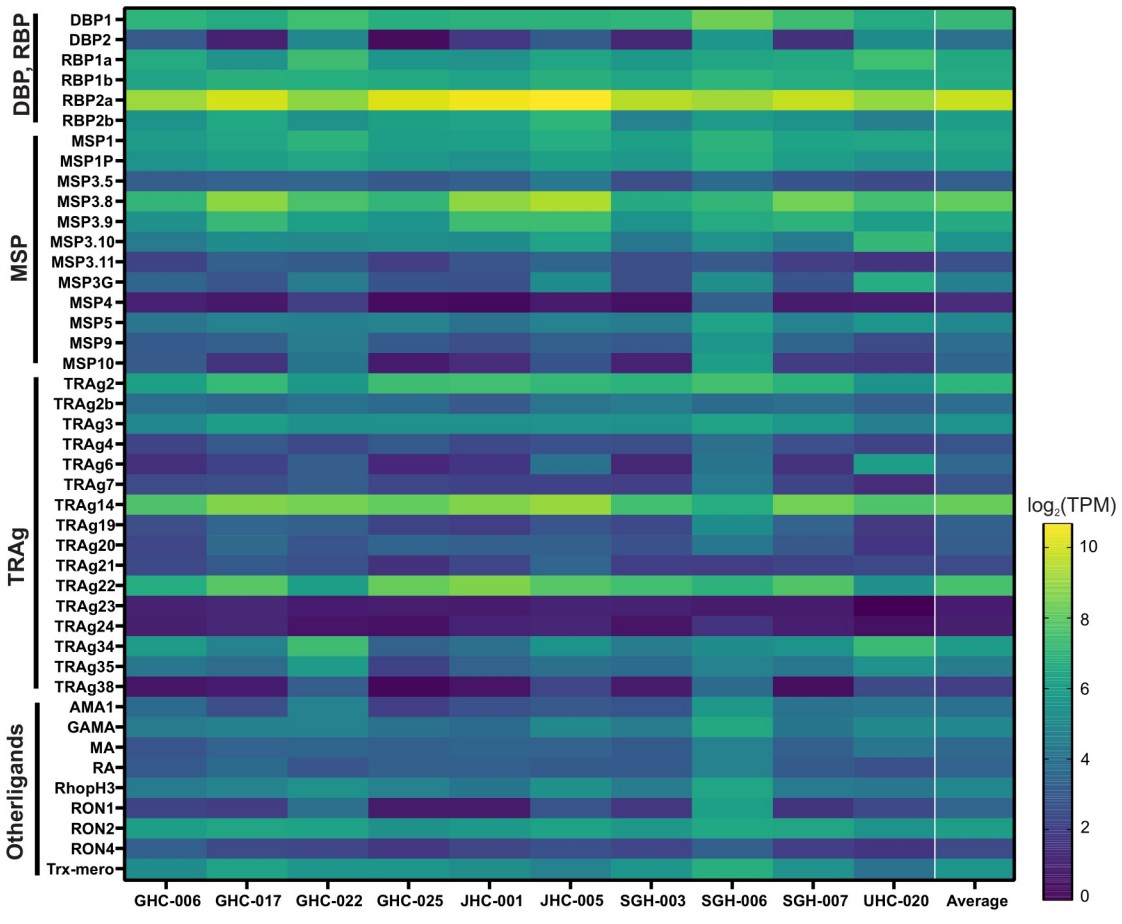

**Ethiopian *P.vivax***

**Fig 5. Heatmap showing 43 genes associated with erythrocyte binding function in the Ethiopian *P. vivax* based on *log*(2)TPM +1 values.** *PvRBP*2b showed four-fold higher expression on average than *PvEBP/DBP*2, but 50% less than *PvDBP*1. *PvRBP*2a showed consistently the highest expression across all samples, with about 6-fold, 67-fold, and 15-fold higher expression than *PvDBP*1, *PvEBP/DBP*2, and *PvRBP*2b, respectively. Other genes including *PvMSP*3.8, *PvTRAg*14, and *PvTRAg*22 also showed higher expression than *PvDBP*1.

and two samples from Brazil (Fig 7). Amongst the 26 gametocyte-related genes, *PvAP2-G* (PVP01_1440800) as well as the gametocyte associated protein, GAP (PVP01_1403000) and *Pvs47* (PVP01_1208000) from female and male gametocytes, respectively, showed the highest expression across the Ethiopian, Cambodian, and Brazilian isolates, and were consistently higher than *Pvs*25 (Fig 7). This expression pattern suggests the potential utility of these three genes as better gametocyte biomarkers across geographical isolates. Other genes indicated differential expression patterns among isolates, e.g., the female gametocyte gene PVP01_0904300 (CPW-WPC family protein) showed consistently high levels of expression in both the Ethiopian and Cambodian isolates, though much lower in the Brazilian ones. On the other hand, PVP01_1302200 (high mobility group protein B1) and PVP01_1262200 (fructose 1,6-bisphosphate aldolase) from the female and male gametocytes showed the highest expression levels in Brazilian *P. vivax* but not the Ethiopian and Cambodian ones.

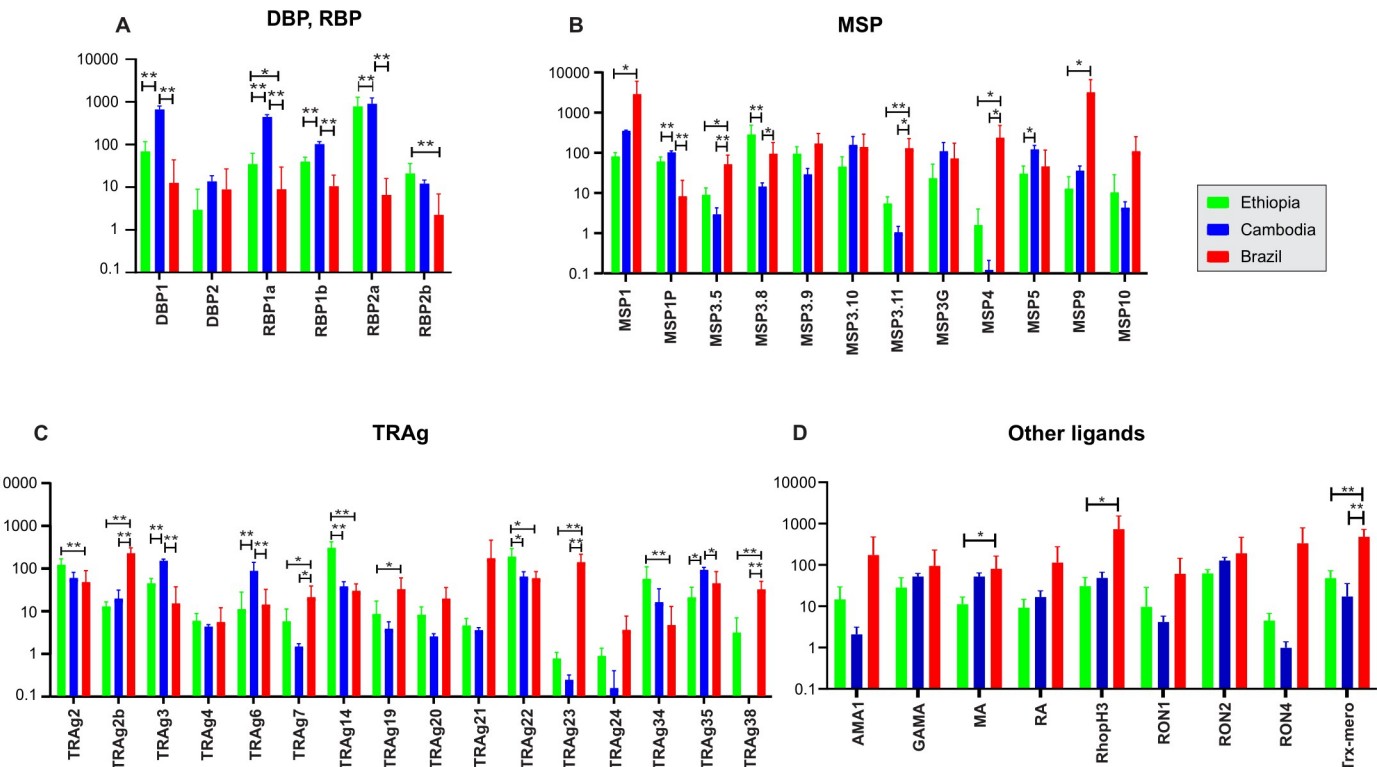

**Fig 6.** Comparisons of 43 genes associated with erythrocyte binding function based on *log*(2)TPM+1 values across the Ethiopian, Cambodian, and Brazilian *P. vivax* for (A) *PvDBP*1, *PvEBP*, and *PvRBP* genes; (B) *PvMSP* genes; (C) *PvTRAg* genes; (D) other putatively functional ligands. * denotes *P*-value < 0.05; ** denote *P*-value <0.01.

## 4. Discussion

This study is the first to examine the transcriptomic profile of *P. vivax* from Africa and compare gene expression among geographical isolates. Approximately 32% of the detected transcripts are of unknown function, some of which such as PVP01_0319500, PVP01_1011500, and PVP01_1228800 were among the highest expressed and could play critical function. It is not surprising that 23% of the highly expressed transcripts belong to housekeeping function, such as several zinc fingers and ATP-synthase proteins. Besides, there is a large number of highly expressed protein regulators and PTMs that have not been thoroughly examined. For example, PVP01_1444000, a ubiquitin-activating enzyme, was among the highest expressed transcripts but with unclear function. Several other protein kinases, lysophospholipases, and chaperones were also highly expressed but their role in intercellular signaling pathways is unclear. It is worth noting that a great proportion of transcripts responsible for ribosomal protein production were highly expressed compared to other gene categories. These ribosomal proteins support intraerythrocytic development of the parasites from one stage to another.

Members of the RBP family including *PvRBP1a*, *PvRBP2a*, *PvRBP2b*, and *PvRBP3* were consistently highly expressed across the Ethiopian and Cambodian but not the Brazilian isolates, suggestive of potential differences in their role of erythrocyte invasion. Recent studies showed that the binding regions of *PvRBP1a* and *PvRBP1b* are homologous to that of *PfRh*4, and the amino acids at site ~339–599 were confirmed to interact with human reticulocytes [57]. Though the host receptors of both PvRBP1a and PvRBP1b proteins are unclear, their receptors are neuraminidase resistant [31]. Recently, transferrin receptor 1 (TfR1) has been

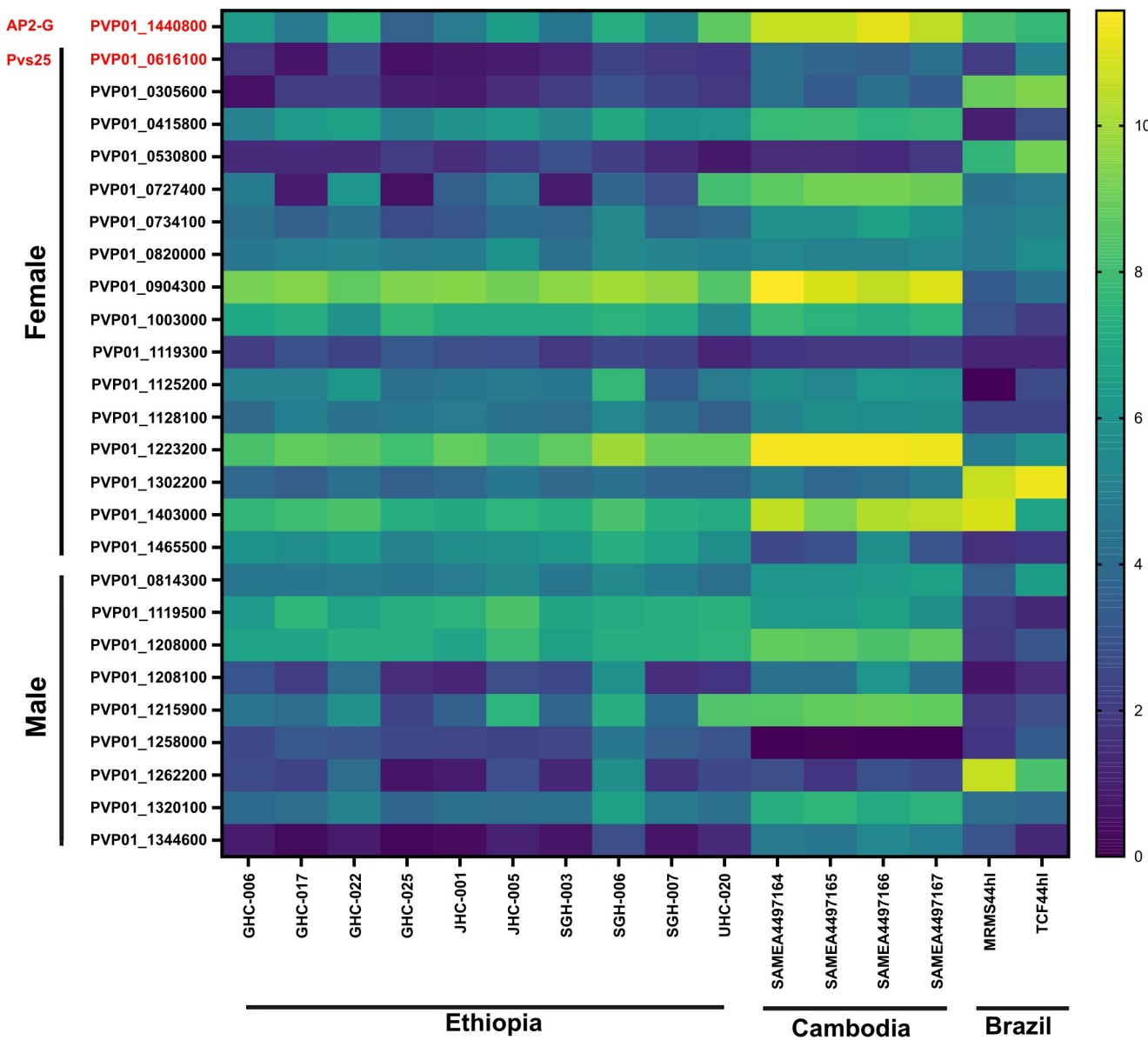

**Fig 7. Heatmap comparing 26 *P. vivax* gametocyte biomarker candidates across the Ethiopian, Cambodian, and Brazilian *P. vivax*.** Based on the expression level of *Pvs25*, all 10 *in vitro P. vivax* samples from Ethiopia, four samples from Cambodia, and two samples from Brazil contained gametocytes. Three genes, PVP01_1440800 (*PvAP2-G*), PVP01_1403000 (gametocyte associated protein, GAP), and PVP01_1208000 (*Pvs47*) from female and male gametocytes, respectively, showed the highest expression across all geographical isolates, and were consistently higher than *Pvs25*.

identified as the receptor for PvRBP2b and the PvRBP2b-TfR1 interaction plays a critical role in reticulocyte invasion in Duffy-positive infections [34]. *PvRBP*2d, *PvRBP*2e, and *PvRBP*3 are pseudogenes that share homology with other *PvRBP*s but encode for nonfunctional proteins [58]. The extent to which of these *PvRBP* genes involve, if any, in erythrocyte invasion remains unclear and requires functional assays in broad samples. The high expression of *PvRBP* genes in Ethiopia could be related to a greater proportion of individuals having low levels of DARC expression (i.e., Duffy-negatives) [3], where *P. vivax* can infect and adapt to both Duffy-positive and Duffy-negative populations [59]; whereas in Cambodia and the inland regions of Brazil, populations are predominantly Duffy-positive [3]. Given that *P. falciparum* can modulate

gene expression in response to their hosts through epigenetic regulation [42,60,61], higher *PvRBP* expression in the Ethiopian *P. vivax* could be a response to the host Duffy phenotype. Further investigation on the expression and binding affinity of these *PvRBP* genes in different Duffy groups is necessary.

Another invasion protein, RIPR, was also among the highly expressed transcripts in *P. vivax*. RIPR is currently known as a vaccine target in *P. falciparum* [62], where RIPR (PfRH5) binds to the erythrocyte receptor basigin [63,64]. The PfRh5 complex is composed of PfRh5, Ripr, CyRPA, and Pf113, which collectively promote successful merozoite invasion of erythrocytes by binding to basigin (BSG, CD147) [64,65]. A BSG variant on erythrocytes, known as Ok$^{a-}$, has been shown to reduce merozoite binding affinities and invasion efficiencies [63], though this has only been reported in individuals of Japanese ancestry [66]. Despite the clear role of RIPR in *P. falciparum*, *P. vivax* RIPR does not seem to bind to BSG [67] and the exact role of RIPR and its binding target(s) remains unclear.

The KR-DE analysis showed 10–26% variation among the transcriptomes of the three countries, with the Ethiopian and Cambodian *P. vivax* being most similar whereas the Cambodian and Brazilian *P. vivax* most different. Genes that showed the highest levels of differentiation were those involved in housekeeping, PIR, and ribosomal functions. The exact reason for such differences amongst the geographical *P. vivax* isolates remains unclear. Earlier whole genome sequencing analyses indicated clear genetic distinction between Southeast Asian and South/Central American *P. vivax* populations [68]. *P. vivax* from East Africa (Madagascar and Mauritania) was closely related to the Indian isolates and intermediate between the clades of Asia and the Americas. More recent study including broader African samples indicated that the Ethiopian, Cambodian, and Brazilian *P. vivax* are independent subpopulations, with isolates from Southeast Asia and East Africa share common ancestry [69]. Microsatellite analyses of global *P. vivax* further showed that the South American *P. vivax* were more related to the Asian populations while the Central American *P. vivax* were more closely related to some African populations [67], suggesting a recent introduction of *P. vivax* from Asia and Africa into America. These genetic relationships may reflect the ancient connections between African and Asian (Old World) *P. vivax* populations and suggest that Asian *P. vivax* populations may have genetically mingled with the American (New World) lineages to a limited extent in recent times and explain variations in the expression profiles. Alternatively, in *P. falciparum*, host nutrition has been shown to significantly alter gene expression related to housekeeping, metabolism, replication, and invasion/transmission [70]. A prior study has shown malnourishment can offer a protective effect to *P. vivax* infections in people from the western Brazilian Amazon [71], though it remains uncertain if this could contribute to genetic relationships observed. In zebra fish, sex determination can cause significant expressional differences in the housekeeping genes [72], suggesting that sexual development factors may alter expression profiles. Technical differences between the study sites, such as cryopreservation and schizont maturation techniques used in Ethiopia and Brazil, or the small presence of ring-stage parasites identified in Brazilian isolates, may alter expression profiles, though to what degree remains uncertain. Future studies with expanded geographical samples are needed to draw more definitive conclusions. Additionally, studies comparing expression profiles of cryopreserved and fresh parasite *ex-vivo* short-term cultures would validate the results of this study.

*P. vivax* PIR genes support a wide range of functions, including antigenic variation, immune evasion, sequestration, and adhesion [73,74]. Gene expression studies suggested their prominent role in virulence and chronic infections. In *P. berghei*, the *pir* transcriptional repertoire is diverse with different members or subfamilies expressed at different time throughout the parasite developmental cycle [75]. PIR proteins have been shown to be targeted by antibodies [76]. The high expression observed for some PIR proteins, such as PVP01_1000200, in the

Cambodian and Ethiopian *P vivax* may suggest the prominent role of VIR antigens in epigenetic regulation associated with host exposure and immune responses [42,60,61], and such immune responses could vary in diverse geographical settings [77–79]. Varying expression of ribosomal proteins, such as PVP01_0827400 (60S ribosomal protein L26) and PVP01_1013900 (40S ribosomal protein S9, putative) may be attributed to host nutrition, which is directly proportional to the speed of replication in *P. berghei* [70]. Future studies should examine host factors associated with the expression of these genes in *P. vivax*.

In this study, the deconvolution of stage-specific transcripts was based on the *P. berghei* orthologues rather than the single-cell RNA-seq data of *P. vivax* because the latter did not show expression from the ring stage. To date, *P. berghei* remains the most comprehensively characterized single-cell data for both sexual and asexual blood stages of *Plasmodium* [80,81], and their orthologues have been shown to be reliable for determining stage-specific transcripts [53]. In primates, most *P vivax* genes have been shown to transcribe during a short period in the intraerythrocytic cycle [82] with a high proportion of late-schizont transcripts expressed as early as the trophozoite stage. In *P. berghei*, the process of gametocyte development and genes involve in sequestration are transcribed much earlier during the trophozoite-schizont transition stage. Male gametocyte development precursors are expressed in the asexual stages prior to the onset of gametocyte development [83,84]. For example, the transcription factor *AP2-G* in *P. vivax* expresses early in the asexual stage for parasites that are committed to sexual development [54]. These factors hinder deconvolution efforts, making it challenging to identify which genes are transcribed in each stage precisely. Future studies should consider combining *in vivo* (rich in ring and trophozoites) and *in vitro* (rich in trophozoites and schizonts) RNA-seq data to provide a more comprehensive and reliable stage-specific model for deconvolution.

Low density *P. vivax* gametocytes in asymptomatic carriers can significantly contribute to transmission [85,86]. In areas with low transmission, submicroscopic infections are hidden reservoirs for parasites with high proportions of infectious gametocytes [87]. The current gametocyte biomarkers *Pvs25* (PVP01_0616100) and *Pvs16* (PVP01_0305600) account only for female gametocytes [88], and grossly underestimate the total gametocyte densities. We previously described two alternative female (PVP01_0415800 and PVP01_0904300) and one male (PVP01_1119500) gametocyte genes that show higher expression than *Pvs25* in the Ethiopian isolates [43]. Nevertheless, these genes showed relatively low expression in the Cambodian and Brazilian isolates. By contrast, *PvAP2-G* (PVP01_1440800), GAP (PVP01_1403000), and *Pvs47* (PVP01_1208000) were moderately expressed across all geographical isolates, and at a level higher than *Pvs25*. Thus, these genes warrant further investigations on their potential utility as gametocyte biomarkers in low-density infections, as well as their exact role in gametocyte development.

## 5. Conclusion

This paper characterized the first *P. vivax* transcriptome from the African isolates and identified several host-interaction gene transcripts, including *PvRBP2*a, *PvMSP*3.8, *PvTRAg*14, and *PvTRAg*22 that were highly expressed compared to *PvDBP1* in parasites from human populations where Duffy negativity is rare or absent. We further demonstrated 10% to 26% differences in the gene expression profile amongst the geographical isolates, with the Ethiopian and Cambodian *P. vivax* being most similar. These findings provide an important baseline for future comparisons of *P. vivax* transcriptomes with Duffy-negative infections. Further investigations examining binding affinity and functionality of *P. vivax* ligands, especially *PvRBP2*a, *PvMSP*3.8, *PvTRAg*14, and *PvTRAg*22 are imperative to clarify their role in erythrocyte invasion. Furthermore, PVP01_1440800 (*PvAP2-G*), PVP01_1403000 (*GAP*), and PVP01_1208000

(*Pvs*47) of both female and male gametocytes showed higher expression than the standard *Pvs25* in all geographical *P. vivax*. These gene markers may provide better gametocyte detection for low-density infections.

## Supporting information

**S1 Table. Name and gene ID of 43 candidate invasion genes.**
(XLSX)

**S2 Table.** Kenward-Roger DE analyses comparing the differentially expressed genes between (a) Ethiopian and Cambodian, (b) Ethiopian and Brazilian, and (c) Cambodian and Brazilian *P. vivax*.
(XLSX)

## Acknowledgments

We thank the field team from Jimma University for their technical assistance; the communities and hospitals for their support and willingness to participate in this research; and undergraduate students at UNC Charlotte for assistance with the experiments.

## Author Contributions

**Conceptualization:** Daniel Kepple, Colby T. Ford, Jean Popovici, Delenasaw Yewhalaw, Eugenia Lo.

**Data curation:** Daniel Kepple, Colby T. Ford, Jonathan Williams, Beka Abagero, Shaoyu Li, Jean Popovici, Delenasaw Yewhalaw, Eugenia Lo.

**Formal analysis:** Daniel Kepple, Colby T. Ford, Jonathan Williams, Beka Abagero, Shaoyu Li, Jean Popovici, Delenasaw Yewhalaw, Eugenia Lo.

**Funding acquisition:** Jean Popovici, Eugenia Lo.

**Methodology:** Daniel Kepple, Colby T. Ford, Jonathan Williams, Beka Abagero, Shaoyu Li, Jean Popovici, Delenasaw Yewhalaw, Eugenia Lo.

**Resources:** Daniel Kepple, Colby T. Ford, Beka Abagero, Shaoyu Li, Jean Popovici, Delenasaw Yewhalaw, Eugenia Lo.

**Supervision:** Eugenia Lo.

**Writing – original draft:** Daniel Kepple, Colby T. Ford, Jonathan Williams, Beka Abagero, Shaoyu Li, Jean Popovici, Delenasaw Yewhalaw, Eugenia Lo.

**Writing – review & editing:** Daniel Kepple, Eugenia Lo.

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
