## [Decision Letter · Decision Letter 0]

6 Dec 2023

Dear Dr. Kepple,

Thank you very much for submitting your manuscript "Comparative transcriptomics reveal differential gene expression among Plasmodium vivax geographical isolates and implications on erythrocyte invasion mechanisms" for consideration at PLOS Neglected Tropical Diseases. As with all papers reviewed by the journal, your manuscript was reviewed by members of the editorial board and by several independent reviewers. The reviewers appreciated the attention to an important topic. Based on the reviews, we are likely to accept this manuscript for publication, providing that you modify the manuscript according to the review recommendations. 

Sincerely,

Ananias A. Escalante, PhD

Academic Editor

Walderez Dutra

Section Editor

Reviewer's Responses to Questions

**Key Review Criteria Required for Acceptance?**

**Methods**

-Are the objectives of the study clearly articulated with a clear testable hypothesis stated?

-Is the study design appropriate to address the stated objectives?

-Is the population clearly described and appropriate for the hypothesis being tested?

-Is the sample size sufficient to ensure adequate power to address the hypothesis being tested?

-Were correct statistical analysis used to support conclusions?

-Are there concerns about ethical or regulatory requirements being met?

Reviewer #1: Objectives and methods are properly described. Statistical analysis seems appropriate. No ehtical concerns were identified.

**Results**

-Does the analysis presented match the analysis plan?

-Are the results clearly and completely presented?

-Are the figures (Tables, Images) of sufficient quality for clarity?

Reviewer #1: Overall, the manuscript is well written and results are prearly presented and discussed.

**Conclusions**

-Are the conclusions supported by the data presented?

-Are the limitations of analysis clearly described?

-Do the authors discuss how these data can be helpful to advance our understanding of the topic under study?

-Is public health relevance addressed?

Reviewer #1: Comparing Plasmodium vivax transcriptomes obtained with different study protocols is necessarily challenging. This authors might perhaps discuss the potential limitations of the comparative approach in greater detail.

**Editorial and Data Presentation Modifications?**

Reviewer #1: None.

**Summary and General Comments**

Reviewer #1: Kepple and colleagues have generated 10 new transcriptomes of Plasmodium vivax isolates from Ethiopia and compare their results with previously published data from Cambodia (4 transcriptomes) and Brazil (2 transcriptomes). 

In two of the studies (Ethiopia and Brazil), cryopreserved, patient-derived isolates were thawed and maintained in short-term culture ex-vivo for approximately 40-44 hours, under slightly different conditions, for schizont maturation; in one study (Cambodia), fresh parasites were used for ex-vivo maturation. The key question is whether differences in transcriptomic profiles among parasites were mostly due to the geographic origin of parasites (Horn of Africa vs. Southeast Asia vs. Amazon Basin in South America) or to differences in sample preparation protocols prior to transcriptomic analysis. This question deserves deeper discussion throughout the manuscript. 

Interestingly, CIBERSORTx deconvolution analysis shows the presence of some ring-stage transcripts in Brazilian isolates (Figure 1), which is unexpected after the ex-vivo maturation process. However, the difference in the proportion of late trophozoites and schizonts among samples of different geographic origin did not reach statistical significance.

Why are transcriptomes of Cambodian and African parasites more similar to each other than they are to those from South American samples is not entirely clear. Population genomic analyses (e.g., doi: 10.1038/ng.3588) typically show greater similarity between African and American lineages of P. vivax , compared with Southeast Asian lineages. Again, can this be due to differences in study protocols used for samples processing and transcriptomic analyses rather than geographic differences in transcription patterns?

Two minor comments: 

1. Because the genes coding for the PIR family in P. vivax had originally been described as the vir multigene family in P. vivax, the nomenclature may be a bit confusing. 

2. The authors coreectly state that "malnourishment has a protective effect to P. vivax infections in people from the western Brazilian Amazon (70)". Yes, there is a single cross-sectional report of such an association in children, with suboptimal controlling for confounders, which cannot be taken as evidence for a causal association between malnutrition and protection from malaria. I suggest these data are not overinterpreted in this context (differences between the transcriptomes of two P. vivax lineages from adults in Brazil and those from other geographic regions) as population-based nutritional status data from the study sites are not considered here.

PLOS authors have the option to publish the peer review history of their article (what does this mean?). If published, this will include your full peer review and any attached files.

Reviewer #1: No

Figure Files:

Data Requirements:

Reproducibility:

References

---

## [Decision Letter · Decision Letter 1]

19 Jan 2024

Dear Dr. Kepple,

We are pleased to inform you that your manuscript 'Comparative transcriptomics reveal differential gene expression among Plasmodium vivax geographical isolates and implications on erythrocyte invasion mechanisms' has been provisionally accepted for publication in PLOS Neglected Tropical Diseases.

Best regards,

Walderez O. Dutra, PhD.

Section Editor

Walderez Dutra

Section Editor

Reviewer's Responses to Questions

**Key Review Criteria Required for Acceptance?**

**Methods**

-Are the objectives of the study clearly articulated with a clear testable hypothesis stated?

-Is the study design appropriate to address the stated objectives?

-Is the population clearly described and appropriate for the hypothesis being tested?

-Is the sample size sufficient to ensure adequate power to address the hypothesis being tested?

-Were correct statistical analysis used to support conclusions?

-Are there concerns about ethical or regulatory requirements being met?

Reviewer #1: (No Response)

**Results**

-Does the analysis presented match the analysis plan?

-Are the results clearly and completely presented?

-Are the figures (Tables, Images) of sufficient quality for clarity?

Reviewer #1: (No Response)

**Conclusions**

-Are the conclusions supported by the data presented?

-Are the limitations of analysis clearly described?

-Do the authors discuss how these data can be helpful to advance our understanding of the topic under study?

-Is public health relevance addressed?

Reviewer #1: (No Response)

**Editorial and Data Presentation Modifications?**

Reviewer #1: (No Response)

**Summary and General Comments**

Reviewer #1: (No Response)

PLOS authors have the option to publish the peer review history of their article (what does this mean?). If published, this will include your full peer review and any attached files.

Reviewer #1: **Yes: **Marcelo U. Ferreira

---

## [Editor Report · Acceptance letter]

25 Jan 2024

Dear Dr. Kepple,

We are delighted to inform you that your manuscript, "Comparative transcriptomics reveal differential gene expression among Plasmodium vivax geographical isolates and implications on erythrocyte invasion mechanisms," has been formally accepted for publication in PLOS Neglected Tropical Diseases.

Best regards,

Shaden Kamhawi

co-Editor-in-Chief

Paul Brindley

co-Editor-in-Chief
